# Current Status of Photodynamic Diagnosis for Gastric Tumors

**DOI:** 10.3390/diagnostics11111967

**Published:** 2021-10-22

**Authors:** Hiroki Kurumi, Tsutomu Kanda, Yuichiro Ikebuchi, Akira Yoshida, Koichiro Kawaguchi, Kazuo Yashima, Hajime Isomoto

**Affiliations:** Department of Multidisciplinary Internal Medicine, Division of Gastroenterology and Nephrology, Faculty of Medicine, Tottori University, 36-1 Nishicho, Yonago 683-8504, Japan; kurumi_1022_1107@yahoo.co.jp (H.K.); tsutomu-k@tottori-u.ac.jp (T.K.); ikebu@tottori-u.ac.jp (Y.I.); akirayoshida1021@yahoo.co.jp (A.Y.); koichiro@tottori-u.ac.jp (K.K.); yashima@tottori-u.ac.jp (K.Y.)

**Keywords:** photodynamic diagnosis, gastric cancer, tumor, endoscopy, photosensitizer

## Abstract

Although the recent development and widespread use of image-enhanced endoscopy and magnifying endoscopy have improved endoscopic diagnosis of gastric cancer, it is somewhat complicated, requires a higher level of expertise, and is still subjective. Photodynamic endoscopic diagnosis (PDED) is based on the fluorescence of photosensitizers that accumulate in tumors, which enables objective evaluation independent of the endoscopist’s experience, and is useful for tumor detection. The objective of this work was to perform a narrative review of PDED for gastric tumors and to introduce our approach to PDED in gastric tumors in our hospital. In our review there have been case reports of PDED for gastric cancer, but its usefulness has not been established because no prospective studies evaluating its usefulness have been performed. In our previous study, 85.7% (42/49) of gastric tumors exhibited fluorescence in PDED. PDED may be useful in the diagnosis of early gastric cancer. Our previous studies were pilot studies in cancer patients; therefore, future prospective studies are required to verify the usefulness of PDED.

## 1. Introduction

According to the latest cancer statistics, 1.09 million new cases of gastric cancer are diagnosed annually worldwide, and 770,000 persons die from gastric cancer annually. Gastric cancer is the sixth most common cancer and the third most common cause of death among all cancer types [1]. Gastric cancer has a 5-year survival rate of over 95% if detected at an early stage [2]. Further, with the recent development and widespread use of endoscopic treatment techniques such as endoscopic submucosal dissection, endoscopic resection can be performed with little or no loss of gastric function if the lesion is detected early enough [3,4]. Therefore, it is important to detect and treat gastric cancer as early as possible.

The recent development and widespread use of image-enhanced endoscopy, such as narrow band imaging (NBI), blue light imaging (BLI), and magnifying endoscopy, have dramatically improved endoscopic diagnosis of gastric cancer [5,6,7,8,9]. In particular, magnified endoscopy with image-enhanced endoscopy, which combines both techniques, has been reported to be useful in the differential diagnosis of gastric cancer and gastritis, as well as in range diagnosis [8,10,11]. However, because the gastric mucosal structure is extremely diverse due to differences in the glandular areas and modification of inflammation associated with *Helicobacter pylori* (*H. pylori*) infection, qualitative diagnosis is subjective and requires the expertise of an endoscopist [10,12]. In contrast, a multicenter study conducted in Japan [13] reported that NBI did not increase the detection rate of early gastric cancer compared to white light observation. A possible explanation for this is that the stomach has a wider lumen than the esophagus and large intestine, and the light intensity of NBI could be inadequate. In addition, with increased *H. pylori* eradication in recent years, there has also been an increase in the number of gastric cancers detected following *H. pylori* eradication. After *H. pylori* eradication, the gastric mucosa exhibits a high incidence of redness. Further, gastric cancer may present gastritis-like findings, thereby making the detection of some gastric cancers after *H. pylori* eradication extremely difficult [11,14,15,16]. Therefore, it is desirable to develop an endoscopic diagnostic method that enables objective evaluation to assist in detecting lesions that is not dependent on the skill of the endoscopist.

Photodynamic diagnosis (PDD) is an optical diagnostic method in which a photosensitizer (PS) or its precursor is administered into the body. Fluorescence is generated by irradiating the PS with excitation light that has accumulated specifically in the tumor. Since PDD visualizes tumors by fluorescence, it is expected to be an objective diagnostic method that is independent of the experience of endoscopists. The objective of this work was to perform a narrative review of photodynamic endoscopic diagnosis (PDED) for gastric tumors and to evaluate the potential of PDED for gastric tumors. We will then introduce our approach to PDED in gastric tumors in our hospital.

## 2. Basic of Photodynamic Endoscopic Diagnosis

PDED utilizes a multiple-stage process involving administration of a PS, selective accumulation of the PS in the target tumor, and illumination by light at the target site. Excitation light and a PS are therefore essential for PDED.

### 2.1. Photosensitizer

The following is a list of PSs that may currently be used for PDED. In 1961, Lipson synthesized hematoporphyrin derivatives (HpD) for localized fluorescence diagnosis of lung and esophageal cancers [17]. In 1979, Dougherty et al. performed photodynamic therapy (PDT) for skin metastases of breast cancer using HpD [18]. This was the first report in which photosensitizers were introduced to treatment on a commercial scale. Subsequently, a mixture of porphyrin dimers and oligomers isolated from HpD was used under the trade name Photofrin^®^ (Concordia Laboratories, Inc., St Michael, Barbados). Photofrin was the first PS to be approved for PDT treatment of bladder cancer in 1993 and is referred to as a first-generation PS. Currently, the United States Food and Drug Administration (US FDA) has approved the use of Photofrin in various types of cancer. Although Photofrin is a potential PS for PDED, it is not the most suitable option because patients experience skin hypersensitivity to light for at least a month due to the long half-life and high accumulation of Photofrin in the skin.

To overcome the disadvantages of Photofrin, the development of second-generation PSs was promoted, including verteporfin (Visudyne^®^; Novartis, Basel, Switzerland), talaporfin sodium (Laserphyrin^®^; Meiji Seika, Tokyo, Japan), and 5-ALA.

Visudyne is a benzoporphyrin derivative of verteporfin, and is a first-line therapy for ocular diseases, age-related macular degeneration, and myopic choroidal neovascularization [19]. For cancer treatment, Visudyne is administered intravenously and becomes most excited by light at a wavelength of approximately 690 nm. The half-life of Visudyne is approximately 5–6 h, and pain, edema, inflammation, bleeding, and discoloration at the injection site after intravenous injection are typical adverse events.

The efficacy of Laserphyrin is well established, especially in PDT [20,21], and is covered by insurance in Japan for esophageal cancer, lung cancer, and malignant brain tumors. Laserphyrin exhibits maximum absorption at a wavelength of approximately 664 nm, which means that it has little effect on the surrounding non-cancerous tissues and is characterized by high penetration into deep tissues. In addition, Laserphyrin is associated with fewer side effects, as a result of its higher selectivity for cancerous tissues and faster elimination of the PS from the body. Compared to Photofrin, which requires light shielding for one month after administration, Laserphyrin only requires two weeks of light shielding.

In vivo, 5-ALA is an amino acid synthesized from glycine and succinyl-coenzyme A. Heme is biosynthesized in vivo through eight enzymatic reactions, including 5-ALA synthesis, and protoporphyrin IX (PpIX) is an intermediate product. Although 5-ALA itself is not a PS, oral administration of 5-ALA causes tumor-specific accumulation of PpIX. PpIX emits red fluorescence when irradiated with 410 nm blue-violet light [22,23]. In 1999, the US FDA approved 5-ALA for use in actinic keratoses of the face and scalp, and it was approved in Europe in 2007 and in the US in 2017 for intracranial use to help in the surgical resection of gliomas. After oral administration of 5-ALA, PpIX fluorescence in the skin peaks between 6 and 10 h. In addition, the blood concentration of 5-ALA reaches its maximum at approximately 30 min after administration and normalizes in approximately 12 h [24]. The 5-ALA is metabolized even faster than Laserphyrin, and the time required for light shielding after administration is generally 48 h. In addition, 5-ALA is more tumor-selective than Photofrin, Visudyne, and Laserphyrin, and is expected to reduce tissue damage (or side effect) to surrounding normal tissues [25].

Based on the above, we believe that 5-ALA is currently the most suitable drug for use in PDED.

### 2.2. Light Source

Light sources used in endoscope systems include xenon, laser, and LED. Currently, the mainstream endoscope systems in Japan are the EVIS LUCERA ELITE series (ELITE) (OLYMPUS Co., Ltd., Tokyo, Japan) equipped with a xenon light source and the LASEREO series (FUJIFILM Co., Ltd., Tokyo, Japan) equipped with a laser light source (FUJIFILM Co., Ltd., Tokyo, Japan).

In ELITE, xenon light passes through the RBG filter and red, green, and blue light are emitted sequentially, and the CCD processes the electrical signals (frame sequential method). ELITE installs a special filter in front of the RBG filter when irradiating light with specific wavelengths such as NBI. A xenon light source with a gentle spectrum with few emission lines is used with a cut filter to focus light of a specific wavelength.

On the other hand, the LASEREO series, which uses an illumination technology that combines two types of semiconductor lasers and phosphors, is equipped with lasers of two different wavelengths. By changing the emission intensity ratio of two lasers, one for WLI with a center wavelength of 450 nm and the other for BLI with a center wavelength of 410 nm, WLI, BLI, BLI-bright, and LCI observation are performed.

We believe that a monophotonic, directional laser could be a suitable light source for PDED because the output light intensity can be adjusted.

## 3. Photodynamic Diagnosis in the Other Fields

PDD have been tried for clinical application in many fields, and their usefulness has been established in some areas.

In the field of neurosurgery, Stummer et al. reported the usefulness of 5-ALA-PDD in glioblastoma removal in 2000 [26], followed by a multicenter Phase III trial on the usefulness of 5-ALA-PDD in malignant glioma surgery in 2006 [27]. There, 5-ALA-PDD significantly improved the rate of total tumor removal (65% vs. 36%) and progression free survival at 6 months (41% vs. 21%) compared to white light. The results of this study led to approval of 5-ALA (Gliolan^®^, photonamic GmbH Co & Medac GmbH.) by the European Medicines Agency in 2007. Since then, the usefulness of 5-ALA-PDD for the resection of malignant gliomas has been reported in many studies. Stummer et al. reported that there was a positive predictive value (PPV) of 100% for strong fluorescence, and a PPV of 92% (biopsy based) or 83% (patient based) for weak fluorescence [28]. Among glioblastoma, similar results were reported in a series of other studies [29,30]. Golub et al. performed a meta-analysis of six studies and found that gross total resection of 5-ALA-PDD for high grade gliomas ranged from 44.7% to 80% and was significantly superior to conventional navigation (OR 2.866, 95% CI 2.127–3.863, *p* < 0.001). They also concluded that 5-ALA-PDD may improve both progression free survival and overall survival [31]. The 5-ALA was approved as an intraoperative diagnostic agent for malignant glioma surgery in Japan in 2013 and by the US FDA in 2017 [32].

PDD with tetracycline and ultraviolet light was reported by Rall et al. in 1957 [33], and Whitmore et al. performed PDD with tetracycline orally and ultraviolet light on 21 patients with bladder cancer in 1964 [34], which was the beginning of PDD in the field of urology. Currently, in the field of urology, 5-ALA-PDD is mainly performed for bladder cancer. The first clinical application of 5-ALA-PDD in bladder cancer was reported by Kriegmair et al. in 1992 using intravesical injection. Further, 5ALA-PDD for bladder cancer increases the detection sensitivity of micro and flat lesions that are difficult to see with white light [35,36,37,38,39]. The detection rate of additional tumors by 5-ALA-PDD is estimated to be 10% to 30%, and the detection rate of carcinoma in situ is especially markedly improved [40,41,42,43,44,45]. The sensitivity of 5-ALA-PDD is reported to be 93% and the detection rate of carcinoma in situ is reported to be improved by 38.3%. On the other hand, 5-ALA-PDD is also used as an adjunct diagnosis during transurethral resection of bladder tumor (TURBT) for non-muscle layer invasive bladder cancer, and some meta-analyses have reported that the recurrence rate is reduced compared with white light observation [46,47,48]. PDD-assisted TURBT is also expected to enable accurate pathological diagnosis and contribute to appropriate subsequent treatment [42].

Other fields are also attempting clinical applications of PDD. For example, 5-ALA-PDD has been applied to the detection of peritoneal dissemination caused by gastric, ovarian, and pancreatic cancers, and has been reported to improve the detection rate by 10% compared to white light [49,50,51,52,53,54]. The application of PDD is being attempted in many fields, and further verification is expected.

## 4. Photodynamic Endoscopic Diagnosis of Gastric Tumors

We identified relevant studies in the literature by searching the databases of Pubmed. The review was not limited in duration but focused on reports on PDD for gastric tumors published in English. The search terms were as follow: (1) (photodynamic diagnosis gastric cancer); (2) (photodynamic diagnosis gastric tumor); (3) (PDD gastric cancer); (4) (PDD gastric tumor). We also read the reference lists of the selected studies to manually identify further relevant studies. Articles were excluded if: (1) The article was a basic research or commentary; (2) the study had insufficient information and descriptions; (3) the article was not published in English; (4) the full text was unavailable.

In 1999, Mayinger et al. performed in vivo PDED using 5-ALA for a 1.5-cm gastric adenoma in an 85-year-old male. Six to seven hours after oral administration of 5-ALA (15 mg/kg), 375–440 nm light produced by the cut filter was irradiated into the adenoma, and red fluorescence was observed [55]. In 2013, Nakamura et al. performed PDED on an 87-year-old patient with gastric cancer 6 h after intravenous administration of talaporfin sodium (40 mg/m^2^) and confirmed clear red fluorescence [56]. Further, Nakamura et al. performed PDD on 10 resected specimens of gastric tumors that were resected 4 h after oral administration of 5-ALA, and confirmed red fluorescence in 60% of the lesions [57]. There were no reviews or meta-analyses that evaluated PDED for gastric tumors.

Thus, PDED is expected to be applied to the endoscopic diagnosis of gastric tumors, but its usefulness has not been established because no prospective studies evaluating its usefulness have been performed.

## 5. PDED for Gastric Tumors at Our Hospital

Firstly, we performed validation using a gastric cancer cell line. When 5-ALA was added to gastric cancer cell lines (MKN-45 and MKN-74) and irradiated with 410 nm excitation light 4 h later, red fluorescence was confirmed (Figure 1).

Subsequently, we developed a prototype endoscope equipment called Sie-P1, which is capable of PDED, and investigated whether fluorescence could be detected in gastric tumors. Sie-P1 is an endoscope system based on the laser endoscope system LESEREO4450 (FUJIFILM Co., Ltd., Tokyo, Japan), and consists of the processor VP-0001 and the light source LL-4450-P1. Sie-P1 is an all-in-one type endoscope with an internal laser light source that can immediately emit excitation light for PDED at any time during endoscopic examination. The white light mode for conventional observation and the excitation light mode for PDED can be easily switched with the push of a single button. We performed PDED on 33 lesions in 30 patients using the Sie-P1. In the PDED method, 5-ALA (20 mg/kg) was dissolved in water and the appropriate dose was orally administered on the day of treatment according to the dosage of 5-ALA preparations for malignant glioma and bladder cancer. After 3–6 h, the gastric tumor was irradiated with laser light to excite PpIX and Sie-P1 was used to check for red fluorescence. To avoid a phototoxic reaction, patients were managed with light <500 L × for 24 h after taking oral 5-ALA. Fluorescent tumors with a border to the surrounding area were considered positive, and 26 of 33 lesions (78.8%) were positive with PDED (Figure 2). However, the overall image was dark, and the fluorescence was weak, which required improvement [58,59].

Therefore, we developed Sie-P2, a prototype endoscope with further technical improvements, such as increasing the laser power and eliminating the cut filter on the tip of the scope to increase the contrast between the background and the tumor area. Sie-P2 is an endoscope system based on LESEREO7000 (FUJIFILM Co., Ltd.) and consists of a processor VP-7000-P2 and a light source LL-7000-P2. Similar to Sie-P1, Sie-P2 can instantly alternate between the excitation light for PDED and white light at the touch of a button. PDED was performed on 16 lesions in 13 patients, and all lesions (100%) were positive with PDED (Figure 3). Furthermore, compared to Sie-P1, the entire image was brighter and more easily seen [60].

Analysis of all gastric tumors in which PDED was performed using Sie-P1 and Sie-P2 showed fluorescence in 42 of 49 lesions in 43 cases (85.7%). The examinations with Sie-P1 and Sie-P2 were associated with adverse events such as elevated white blood cell count (14.0% [6/43]), nausea (2.3% [1/43]), and aspartate aminotransferase elevation (2.3% [1/43]), but careful follow up revealed that these events resolved spontaneously.

We compared the clinicopathological features (age, sex, tumor size, histopathology, tumor depth, and macroscopic type) of the 42 lesions in the PDED-positive group in which tumor fluorescence could be confirmed and the seven lesions in the PDED-negative group in which fluorescence could not be confirmed (Table 1). There was a significant difference in the histopathological type between the PDED-positive and PDED-negative groups, particularly in the case of signet ring cell carcinoma, in which PDED was negative in all cases. We hypothesized that the heme biosynthetic pathway—a metabolic pathway of 5-ALA—may differ depending on the histopathological type of the tumor. Considering PpIX is a metabolite produced by the heme biosynthetic pathway, it is reasonable to assume that the accumulation of PpIX in tumor cells after 5-ALA administration is due to changes in the expression or activity of metabolic enzymes and transporters related to the heme biosynthetic pathway in tumor cells. We examined the expression of enzymes and transporters related to the heme biosynthetic pathway using immunohistochemical staining of resected specimens (Figure 4). We found that coproporphyrinogen-III oxidase, which catalyzes the reaction of coproporphyrinogen III to protoporphyrinogen, and protoporphyrinogen oxidase, which catalyzes the reaction of protoporphyrinogen to PpIX, were downregulated in the signet-ring cell carcinomas (Figure 5) [58,59].

## 6. Future Prospects

In the prior studies, 85.7% (42/49) of gastric tumors exhibited fluorescence in PDED [58,59,60]. PDED is based on the fluorescence of the tumor, which enables objective evaluation independent of the endoscopist’s experience, and is useful for tumor detection. Moreover, in these studies, there were gastric cancers were not detected in white light but were detected using PDED. PDED has the potential to improve the diagnostic accuracy of early gastric cancer. However, the existence of gastric cancers that are PDED-negative is a problem that needs to be addressed. In the prior studies, we speculated that the differences in fluorescence were due to differences in the expression of enzymes and transporters involved in the heme biosynthetic pathway associated with histopathological type [58,59]. On the other hand, in the prior study, fluorescence could not be confirmed in all cases of signet ring cell carcinoma. This may be due to the excitation light used for PDED. In other words, we used an excitation light near 410 nm, which is the peak absorption wavelength of PpIX, but due to its short wavelength, it was attenuated in the superficial layer of the mucosa, and fluorescence could not be confirmed in signet ring cell carcinoma arising from the deep layer of the mucosa. Further studies to elucidate the mechanism of tumor specific PpIX accumulation and the optimal excitation wavelength for PDED of early gastric cancer will contribute to the development of PDED for gastric cancer.

On the other hand, with the development of the Sie-P2, despite some improvements, the images are still generally dark and the fluorescence is not good enough. Recently, new endoscope systems equipped with LED light sources, ELUXEO (FUJIFILM Co., Ltd., Tokyo, Japan) were launched in Japan. In the ELUXEO, each LED light with four different wavelengths is optically combined and the light intensity is independently controlled to create an irradiation light with an emission ratio suitable for the situation. LED have a higher light intensity and may be able to produce excitation light more suitable for PDED (Figure 6).

Our previous studies were pilot studies in cancer patients; therefore, future prospective studies are required to verify the usefulness of PDED. In addition, the diagnostic efficacy of PDED for gastric cancer needs to be prospectively compared with WLI, NBI and BLI. Zhou et al. examined the diagnostic efficacy of NBI and BLI in detecting gastric cancer in a meta-analysis of six BLI and 22 NBI reports. The pooled sensitivity of BLI for gastric cancer was 0.89, and the specificity was 0.92. The pooled sensitivity of NBI for gastric cancer was 0.83 and the specificity was 0.95 [9]. We also need to examine the add-on effect of PDED on the diagnostic efficacy of NBI and BLI for early gastric cancer.

In conclusion, we believe that with further validation, PDED will become a useful diagnostic method for gastric cancer.

## Figures and Tables

**Figure 1 diagnostics-11-01967-f001:**
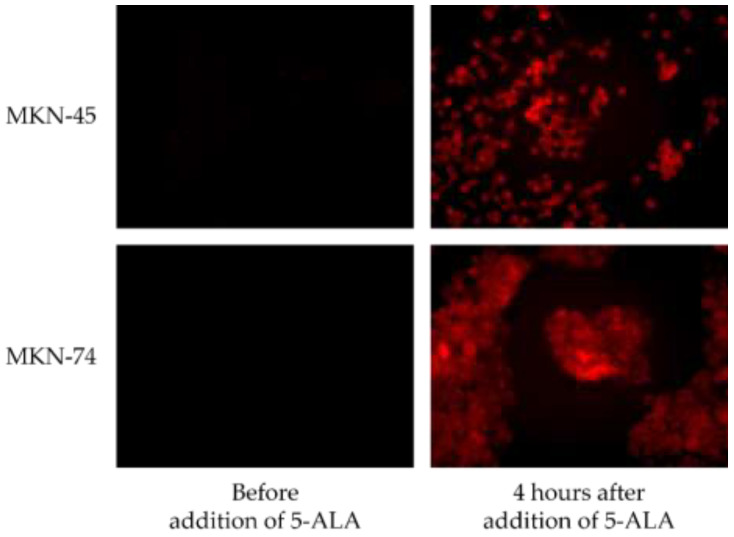
Gastric cancer cell lines supplemented with 5-ALA and irradiated with 410 nm excitation light.

**Figure 2 diagnostics-11-01967-f002:**
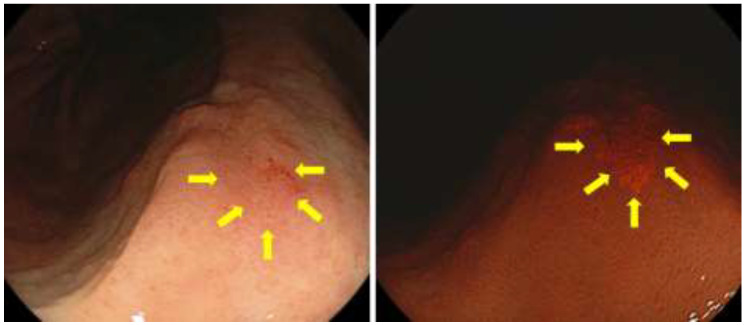
The 5-ALA mediated PDED with Sie-P1 for early gastric cancer.

**Figure 3 diagnostics-11-01967-f003:**
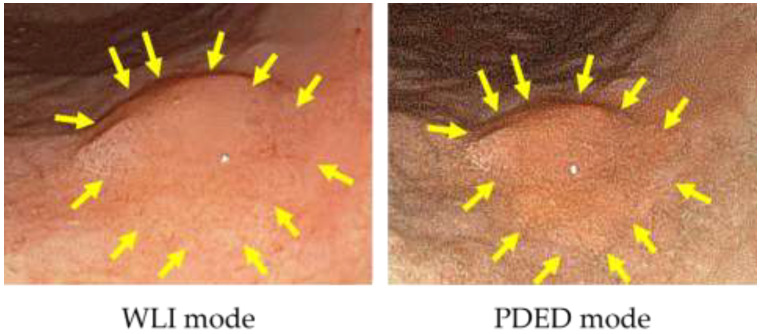
The 5-ALA mediated PDED with Sie-P2 for early gastric cancer.

**Figure 4 diagnostics-11-01967-f004:**
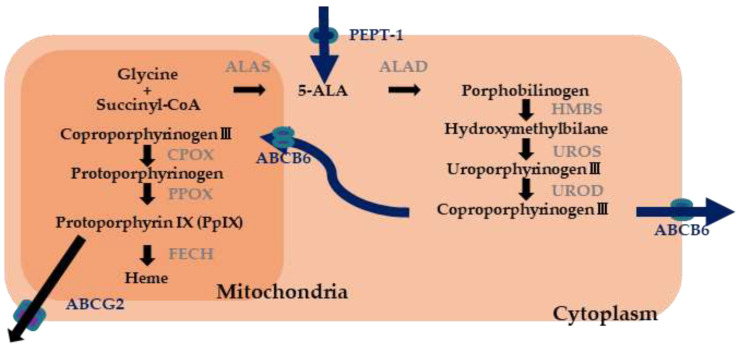
Hemo biosynthetic pathway.

**Figure 5 diagnostics-11-01967-f005:**
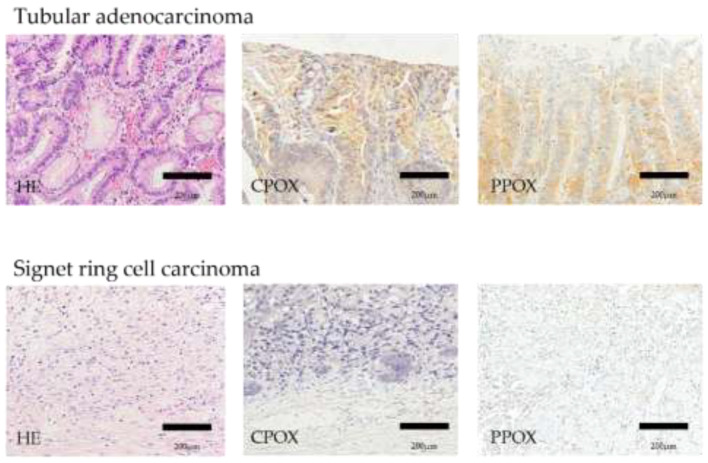
Compared with well differentiated tubular adenocarcinoma, the expression of CPOX and PPOX is decreased in signet ring cell carcinoma.

**Figure 6 diagnostics-11-01967-f006:**
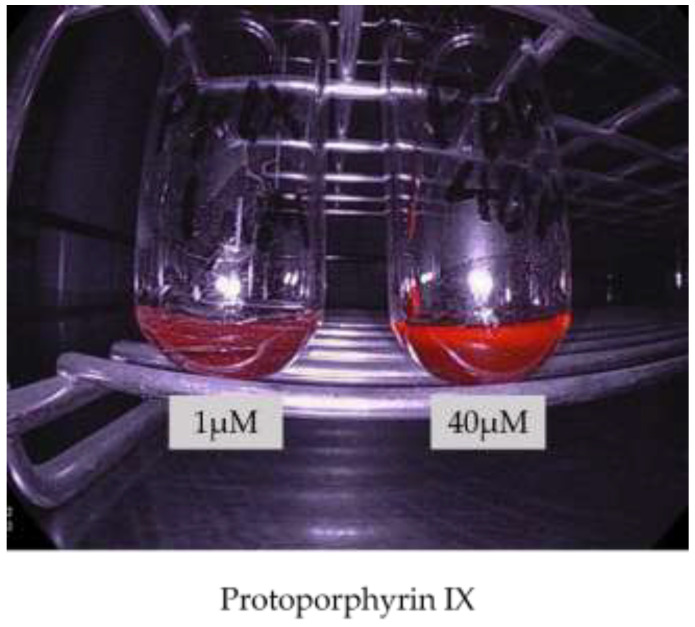
Image of PpIX irradiated with excitation light using ELUXEO equipped with an LED light source.

**Table 1 diagnostics-11-01967-t001:** Comparison of clinicopathological features between PDED positive and negative groups.

	PDED Positive (*n* = 42)	PDED Negative (*n* = 7)	Univariate Analysis	Multivariate Analysis
Sex (male/female)	28/8	4/3	0.87 *	
Age (median)	74.5	64	0.42 **	
Site of tumor (Upper/middle/lower)	4/25/13	0/4/3	0.25 *	
Macroscopic (elevated/flat · depressed)	22/20	0/7	0.01 *	0.42
Tumor diameter (median)	25	7	<0.01 **	0.06
Invasion depth (intramucosal/submucosal)	38/4	6/1	0.75 *	
Histopathology (tub ^1)^/sig ^2)^/adenoma)	33/0/9	3/4/0	<0.01 *	<0.01

* *p* value: Chi square test ** *p* value: *t* test. ^1)^ Tubular adenocarcinoma, ^2)^ Signet-ring cell carcinoma.

## Data Availability

Data sharing is not applicable to this article.

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
