# Peer review of "Current Status of Photodynamic Diagnosis for Gastric Tumors"

_diagnostics, 2021, doi:10.3390/diagnostics11111967_

Round 1
Reviewer 1 Report
In this work, Isomoto and co-workers review the use of photodynamic endoscopic diagnosis (PDED) for the diagnosis of gastric cancer. This review is relatively well-written and such topic would be a good fit for this journal.
However, there are aspects that in my opinion makes this paper non-publishable in this current version. Namely:
-> On the abstract, the authors indicate that PDED is based on the fluorescence of the tumor. However, that it is not correct. PDED is based on the fluorescence of a photosensitizer, which can be present (or not) on the tumor;
-> Basically, this review paper only summarize six papers on the subject of PDED of gastric cancer. To me, this is too short for validating the necessity of a review paper.
-> I've some doubts about the basis of PDED. Namely, the fluorescent probes mentioned by the authors are in fact photosensitizers that produce ROS upon photo-excitation, leading to cell destruction. Given this, how can this approach be used solely for diagnostic, without leading to cell destruction?
-> Following in this topic, PDED appears to be a consequence of photodynamic therapy (PDT) of target tumor tissues. However, if I'm correct, PDT requires previous knowledge about the localization of the tumor, as unwanted photo-activation of the photosensitizer in healthy tissue could lead to tissue damage (and side-effects). Thus, how can PDED be really used to identify a given tissue as being cancerous or not, without side-effects?
-> The type of photosensitizers used in PDT have some difficulties dealing with more deep-seated tumors, due to poor light-penetration into biologic tissues. The authors should discuss if PDED could suffer from the same problem.
Author Response
Dear Reviewer
Thank you very much for your suggestions. We have addressed your comments, and we feel that the manuscript has now been greatly improved as a result. Please see the revised manuscript and confirm our corrections.
On the abstract, the authors indicate that PDED is based on the fluorescence of the tumor. However, that it is not correct. PDED is based on the fluorescence of a photosensitizer, which can be present (or not) on the tumor;
Thank you very much for your valuable comments.
We have revised the manuscript as you suggested.
Page 1, Line 11
Photodynamic endoscopic diagnosis (PDED) is based on the fluorescence of photosensitizers that accumulate in tumors, which enables objective evaluation independent of the endoscopist's experience, and is useful for tumor detection.
Basically, this review paper only summarize six papers on the subject of PDED of gastric cancer. To me, this is too short for validating the necessity of a review paper.
Thank you very much for your very important comments. As the reviewer mentioned, there are currently very few reports that have prospectively validated PDED for gastric cancer. Therefore, we decided to make this report in the form of a narrative review, and I also mentioned possible drugs and light sources that could be used for PDED in gastric cancer. We believe that these information would be beneficial to our readers.
I've some doubts about the basis of PDED. Namely, the fluorescent probes mentioned by the authors are in fact photosensitizers that produce ROS upon photoexcitation, leading to cell destruction. Given this, how can this approach be used solely for diagnostic, without leading to cell destruction?
Thank you very much for your valuable comments.
As the reviewer mentioned, PDED results in the generation of ROS, which may lead to cell destruction. Therefore, a photosensitizer that accumulates more specifically in tumors is desired, and we believe that 5-ALA is the most suitable at this time.
In fact, in bladder cancer and glioma, where PDD with 5-ALA has already been clinically applied, adverse events due to cell destruction have rarely been reported. Since this is an important comment, we have added the following wording.
Page 3, Line 107
In addition, 5-ALA is more tumor-selective than Photofrin, Visudyne, and Laserphyrin, and is expected to reduce tissue damage (or side effect) to surrounding normal tissues.
Following in this topic, PDED appears to be a consequence of photodynamic therapy (PDT) of target tumor tissues. However, if I'm correct, PDT requires previous knowledge about the localization of the tumor, as unwanted photoactivation of the photosensitizer in healthy tissue could lead to tissue damage (and side-effects). Thus, how can PDED be really used to identify a given tissue as being cancerous or not, without side-effects?
Thank you very much for your very important comments!
Thank you very much for your very important comments. As mentioned above, a more tumor-selective photosensitizer is needed to eliminate damage to surrounding normal tissues, and we believe that 5-ALA is currently the most suitable.
Although there are no reports of 5-ALA-PDT in the human stomach, Loh et al [Br J Cancer 1992, 66, 452-462] have confirmed that 5-ALA-PDT causes tissue damage in a tumor-selective manner using a rat model. This is a very important comment and we have added the following wording.
Page 3, Line 107
In addition, 5-ALA is more tumor-selective than Photofrin, Visudyne, and Laserphyrin, and is expected to reduce tissue damage (or side effect) to surrounding normal tissues.
The type of photosensitizers used in PDT have some difficulties dealing with more deep-seated tumors, due to poor light-penetration into biologic tissues. The authors should discuss if PDED could suffer from the same problem.
Thank you very much for your very valuable comments.
We believe that the excitation wavelength around 410 nm, which is also used in our study, provides the strongest fluorescence because it matches the peak of the absorption spectrum of PpIx. However, the short wavelength of 410 nm has low tissue permeability and is strongly attenuated in the mucosal surface layer. Since most early gastric cancers originate from the mucosal surface layer, the short wavelength of 410 nm is often sufficient. However, for cancers arising from the deep mucosa, such as signet ring cell carcinoma, excitation light around 410 nm may not be sufficient due to the effect of light attenuation by the mucosa. We consider it a future challenge to investigate the optimal wavelength of excitation light for the detection of such cancers.
Thank you very much for your valuable comments. We have added the following text to the "Discussion" section.
Page 8, Line 288
On the other hand, in the prior study, fluorescence could not be confirmed in all cases of signet ring cell carcinoma. This may be due to the excitation light used for PDED. In other words, we used an excitation light near 410 nm, which is the peak absorption wavelength of PpIX, but due to its short wavelength, it was attenuated in the superficial layer of the mucosa, and fluorescence could not be confirmed in signet ring cell carcinoma arising from the deep layer of the mucosa.
Further studies to elucidate the mechanism of tumor specific PpIX accumulation and the optimal excitation wavelength for PDED of early gastric cancer will contribute to the development of PDED for gastric cancer.
We would like to thank the reviewer for their helpful comments. I hope that you will find the revised manuscript suitable for publication in Diagnostics.
Sincerely,
Hiroki Kurumi, MD, PhD
Division of Gastroenterology and Nephrology, Department of Multidisciplinary Internal Medicine, Tottori University Faculty of Medicine
36-1, Nishi-cho, Yonago-city, Tottori 683-8504, Japan
E-mail: kurumi_1022_1107@yahoo.co.jp
Phone No: +81-85-938-6527, Fax No: +81-85-938-6529
Reviewer 2 Report
Excellent and very timely review covering the current status of photodynamics role in diagnosing gastric tumors, fully deserving publication.
Author Response
Dear Reviewer
Excellent and very timely review covering the current status of photodynamics role in diagnosing gastric tumors, fully deserving publication.
Thank you very much for your very kind comments. Your support is sincerely appreciated.
We would like to thank the reviewer for their helpful comments. I hope that you will find the revised manuscript suitable for publication in Diagnostics.
Sincerely,
Hiroki Kurumi, MD, PhD
Division of Gastroenterology and Nephrology, Department of Multidisciplinary Internal Medicine, Tottori University Faculty of Medicine
36-1, Nishi-cho, Yonago-city, Tottori 683-8504, Japan
E-mail: kurumi_1022_1107@yahoo.co.jp
Phone No: +81-85-938-6527, Fax No: +81-85-938-6529
Reviewer 3 Report
The article presents a narrative review of photodynamic endoscopic diagnostic (PDED) for gastric tumors and introduces the author's approach to PDED in gastric tumors.
The article is presented in a suitable manner for this kind of study. Sufficient detail is provided and the discussions are based on actual facts and figures. The conclusion is supported by the data, and all the references cited are relevant and adequate.
The article concludes that, with further validation, PDED can become a useful diagnostic method for gastric cancer.
My assessment is that the overall quality of this article is excellent and should be accepted for publishing.
Author Response
Dear Reviewer
The article presents a narrative review of photodynamic endoscopic diagnostic (PDED) for gastric tumors and introduces the author's approach to PDED in gastric tumors.
The article is presented in a suitable manner for this kind of study. Sufficient detail is provided and the discussions are based on actual facts and figures. The conclusion is supported by the data, and all the references cited are relevant and adequate.
The article concludes that, with further validation, PDED can become a useful diagnostic method for gastric cancer.
My assessment is that the overall quality of this article is excellent and should be accepted for publishing.
Thank you very much for your very kind comments. Your support is sincerely appreciated.
We would like to thank the reviewer for their helpful comments. I hope that you will find the revised manuscript suitable for publication in Diagnostics.
Sincerely,
Hiroki Kurumi, MD, PhD
Division of Gastroenterology and Nephrology, Department of Multidisciplinary Internal Medicine, Tottori University Faculty of Medicine
36-1, Nishi-cho, Yonago-city, Tottori 683-8504, Japan
E-mail: kurumi_1022_1107@yahoo.co.jp
Phone No: +81-85-938-6527, Fax No: +81-85-938-6529
Round 2
Reviewer 1 Report
I appreciate the efforts made by authors, but I still think that this manuscript is not suitable for publication. Namely:
- The field is still too narrow to need a review paper (only six reviewed papers);
- The authors still didn't explained how this approach could identify a tumor, if tumor location is previously not accounted for;
- The authors still did not properly explain how this can be a suitable, and solely, diagnostic tool if the probes are essentially cytotoxic.